

# Modelling the growth of the brown frog (*Rana dybowskii*)

Qing Tong[1,2], Xiao-peng Du[3], Zong-fu Hu[3], Li-yong Cui[2] and Hong-bin Wang[1]

[1] College of Veterinary Medicine, Northeast Agricultural University, Harbin, China
[2] Hejiang Forestry Research Institute of Heilongjiang Province, Jiamusi, China
[3] Northeast Agricultural University, Harbin, China

## ABSTRACT

Well-controlled development leads to uniform body size and a better growth rate; therefore, the ability to determine the growth rate of frogs and their period of sexual maturity is essential for producing healthy, high-quality descendant frogs. To establish a working model that can best predict the growth performance of frogs, the present study examined the growth of one-year-old and two-year-old brown frogs (*Rana dybowskii*) from metamorphosis to hibernation (18 weeks) and out-hibernation to hibernation (20 weeks) under the same environmental conditions. Brown frog growth was studied and mathematically modelled using various nonlinear, linear, and polynomial functions. The model input values were statistically evaluated using parameters such as the Akaike's information criterion. The body weight/size ratio ($K_{wl}$) and Fulton's condition factor ($K$) were used to compare the weight and size of groups of frogs during the growth period. The results showed that the third- and fourth-order polynomial models provided the most consistent predictions of body weight for age 1 and age 2 brown frogs, respectively. Both the Gompertz and third-order polynomial models yielded similarly adequate results for the body size of age 1 brown frogs, while the Janoschek model produced a similarly adequate result for the body size of age 2 brown frogs. The Brody and Janoschek models yielded the highest and lowest estimates of asymptotic weight, respectively, for the body weights of all frogs. The $K_{wl}$ value of all frogs increased from 0.40 to 3.18. The $K$ value of age 1 frogs decreased from 23.81 to 9.45 in the first four weeks. The $K$ value of age 2 frogs remained close to 10. Graphically, a sigmoidal trend was observed for body weight and body size with increasing age. The results of this study will be useful not only for amphibian research but also for frog farming management strategies and decisions.

Corresponding author
Hong-bin Wang,
hbwang1940@163.com

## INTRODUCTION

Growth is one of the most important features of animals and can be defined as an increase in body weight or body dimensions over time or with age (*Hossein-Zadeh, 2015*). The growth of frogs is often rapid before maturity but much slower after maturity because more resources are allocated to reproduction (*Halliday & Verrell, 1988*); adult female body length is often positively correlated with fecundity (*Liao, Liu & Merilä, 2015*;

*Liao & Lu, 2009b*). However, male and female frogs show similar trends in growth, body length, and age as the activity period decreases (*Liao & Lu, 2012*).

Currently, studies of amphibian development are mostly based on years (*Liao, Lu & Jehle, 2014*; *Martof, 1956*; *Miaud, Guyétant & Elmberg, 1999*). Several studies have mainly focused on the growth of captive bullfrogs (*Mansano et al., 2012*, *2017a*; *Pereira et al., 2014*). However, the study periods, which mainly involved the tadpole period, and the fattening periods of the bullfrog were relatively short. The growth records of frogs, especially those related to body size, based on weeks or days are very rare, especially for the stages from metamorphosis to sexual maturity. During the stages from metamorphosis to sexual maturity, which are important stages in the life of a frog, larval body length, the duration of the activity period, resource availability, and energy storage vary with different growth patterns (*Bruce, 1993*; *Miaud, Guyétant & Elmberg, 1999*).

Mathematical modelling is used in livestock production to assist technicians and researchers in the development of animal breeding and nutrition programmes to improve the accuracy, profitability, and sustainability of zootechnical activities (*Pereira et al., 2014*). The growth performance of organisms under commercial aquaculture has been a determinant of economic growth, so understanding the factors that limit the growth performance of brown frogs is essential. The literature on frogs and other animals traditionally defines the relationship between age and live weight as a nonlinear, sigmoidal (S-shaped) function (*Kyriakopoulou-Sklavounou, Stylianou & Tsiora, 2008*; *Liao & Lu, 2009a*; *Sarasola-Puente et al., 2011*). However, our preliminary work showed that logistic, Gompertz, and Von Bertalanffy models also fit experimental growth data well (*Qing et al., 2010*), which led us to compare the ability of Handy growth curves, such as second-, third-, and fourth-order polynomial functions, to model the growth of brown frogs with that of polynomial functions. This is an important topic of research because various types of models have different advantages and disadvantages. Polynomial models are much easier to handle than nonlinear models because they are linear from a statistical perspective and can be easily solved using linear regression if certain assumptions are valid. However, nonlinear mathematical functions have been applied extensively in different species to describe body weight development, allowing information from multiple measurements to be combined into a few parameters with biological indications to facilitate both interpretation and comprehension of the phenomenon (*Ersoy, Mendes & Aktan, 2006*; *Hossein-Zadeh, 2015*). Finally, a linear model (first-order polynomial) is especially attractive because of its easy interpretability (*Tompić et al., 2011*).

The brown frog (*Rana dybowskii*) is mainly distributed in north–eastern China, and it is an important species with both medicinal and economic value (*Hu et al., 2016*). After years of research, we have developed a thorough understanding of the conditions necessary for *R. dybowskii* growth and have gained considerable experience in improving breeding conditions. In the present study, we built a suitable growth environment for *R. dybowskii* and investigated its growth in this habitat.

The purpose of this research is to explore the growth performance of brown frogs produced from a frog breeding system. Frog growth was studied and mathematically

modelled using various nonlinear, linear, and polynomial functions. The model input values were statistically evaluated using strategies such as the least squares method, and correlations between the growth curves and the experimental data were analysed by the root mean square error (RMSE), the Durbin–Watson statistic (DW), the Bayesian information criterion (BIC), and Akaike's information criterion (AIC). The use of several nonlinear, linear, and polynomial functions to model frog body weight and size data in this research reflects the most comprehensive study of amphibians to date.

## MATERIALS AND METHODS

### Ethics statement

All frogs used in this study were handled in strict accordance with Northeast Agricultural University (NEAU) Institutional Animal Care and Use Committee (IACUC) protocols (IACUC#09-012) and tissues-of-opportunity waivers were approved by NEAU. Tissues-of-opportunity are defined as samples collected (1) during the course of another project with an approved IACUC protocol from another institution; (2) during normal veterinary care provided by appropriately permitted facilities; or (3) from free-ranging animals at appropriately permitted facilities. No endangered or protected species were harmed during this study.

### Breeding environment

The experiments were carried out from April to October of 2010 at a farm located in Jiankou Town (46°51′54″N, 130°17′32″E; 80 m above sea level) owned by the Folin company in Jiamusi City, Heilongjiang Province, China. The area experiences a temperate continental monsoon climate with a long winter and a short summer, a frost-free season that typically lasts for 130 d, average yearly precipitation of approximately 510 mm, and an average temperature of 2.8 °C.

The open-air breeding pens were enclosed by a sealed fence, a sprinkler was installed, and some low vegetation was planted. This simulated environment designed for rearing brown frogs resembled an actual forest habitat to ensure the necessary conditions for growth. Age 1 and age 2 brown frogs were reared separately in different pens, but all pens were maintained under the same conditions, including the air temperature, photoperiod, and air moisture as well as the construction materials and physical properties of the pen. The temperature and humidity were recorded every 2 h by temperature–humidity metres. The air humidity in the pens varied slightly but was maintained at approximately 60~80% by spraying water into the air, and the ground-level humidity of the pens was controlled within a range of 25~35%. During the trial, the average diurnal, maximum, and minimum temperatures recorded inside the facility were 18.73 ± 3.73 °C, 24.36 ± 3.40 °C, and 13.10 ± 2.29 °C, respectively. The temperature of the pens rarely exceeded 35 °C due to ventilation and spraying, thus preventing stress from rapid changes in the frogs' body temperature.

The age 1 and age 2 frogs were reared in three different breeding pens, each of which was approximately 80 m$^2$ in size. The density of the age 1 brown frogs in the pens was 8/m$^2$, and that of the age 2 brown frogs was 4/m$^2$. During the experiments, the frogs and their behaviour were periodically examined. When the average temperature was

above 10 °C and the frogs were observed to be moving, we provided food (*Tenebrio molitor*). The frogs were fed once a day at 09:00, and the feeding amount was approximately 4% of their mean body weight. Insects in the natural environment were also available for consumption. The age 2 frogs used in this experiment had been cultured the previous year (2009) using breeding methods consistent with this experiment (2010). In the fall of 2009, the groups of age 2 frogs were placed in separate wintering ponds to avoid errors in discriminating between the different ages. The initial body weights for the age 1 and age 2 brown frogs were 3.47 ± 0.74 and 0.48 ± 0.05 g, respectively.

## Sample collection

The snout–vent lengths (SVLs) and body weights of 40 brown frogs were measured each week. Callipers with a precision of approximately 0.02 mm were used to measure the body sizes (SVLs) of the specimens, and their body weights were measured using electronic scales (accurate to 0.01 g). Age 1 brown frogs included frogs in the developmental stages from metamorphosis to out-hibernation in the second year, and age 2 brown frogs included frogs in the stages from out-hibernation in the second year to out-hibernation in the third year. Additionally, the one-year-old brown frogs completed their metamorphosis by 5th June and began eating between 10th June and 15th June. Brown frogs cannot eat after 1st October due to low temperatures, which terminates growth, so they must enter hibernation one month later. For age 2 brown frogs, the hibernation period ended on 15th May, and they began to feed from 25th May to 1st June before entering hibernation at the same time as the one-year-old brown frogs.

Males and females were mix-reared in the first year because the brown frogs did not exhibit sexual dimorphism, but late in the second year, males, and females were reared separately.

## Growth functions

Frog growth was studied and mathematically modelled using various nonlinear, linear, and polynomial functions. The nonlinear mathematical models included the logistic, Gompertz, von Bertalanffy, Brody, Janoschek, and Richards models.

These models contain several common parameters and can associate any biological meaning to each of them.

$W$ is the measurement value (g or cm); $t$ is the number of experimental weeks; $A$ is the body weight or length at maturity, $B$ is an integration constant related to the initial weight of the animal, and $C$ is the maturation rate. The value $m$ is the parameter that gives shape to the curve by indicating where the inflection point occurs.

Logistic Model. The following equation describes the Logistic (*Fekedulegn, Mac Siurtain & Colbert, 1999*) growth model:

$$W = \frac{A}{(1 + Be^{(-Ct)})} \tag{1}$$

Gompertz Model. The following equation describes the Gompertz (*Wellock, Emmans & Kyriazakis, 2004*) growth model:

$$W = Ae^{-b\,\exp(-Ct)} \tag{2}$$

Von Bertalanffy Model. The following equation describes the Von Bertalanffy (*Sarasola-Puente et al., 2011*) growth model:

$$W = A(1 - Be^{-Ct})^3 \tag{3}$$

Brody Model. The following equation describes the Brody (*Hossein-Zadeh, 2015*) growth model:

$$W = A\left(1 - Be^{(-Ct)}\right) \tag{4}$$

Janoschek Model. The following equation describes the Janoschek (*Gille & Salomon, 1995*) growth model:

$$W = A - (A - W_0)e^{(-Ct^m)} \tag{5}$$

Richards Model. The following equation describes the Richards (*Fekedulegn, Mac Siurtain & Colbert, 1999*) growth model:

$$W = \frac{A}{(1 + Be^{(-Ct)})^{\frac{1}{m}}} \tag{6}$$

To estimate body weight and body size at a certain age, first-, second-, third-, and fourth-order polynomial functions were fitted to the brown frog body weight and body size data.

Polynomial Model. The following equation describes the Polynomial (*Hadeler, 1974*) growth model:

$$W = d_0 + \sum_{i=1}^{r} d_i \times t^i \tag{7}$$

The value of $r$ is the second to the fourth order of fit; $d_0$ is intercept; $d_i$ is the regression coefficient.

## Prediction consistency and quality

The models were assayed for prediction consistency and quality using an adjusted coefficient of determination ($R^2_{\text{adj}}$). Furthermore, the residual standard deviation, or the RMSE, DW, AIC, and BIC were used.

$R^2_{\text{adj}}$ was calculated as follows:

$$R^2_{\text{adj}} = 1 - \left[\frac{n-1}{n-p}\right](1 - R^2) \tag{8}$$

where $R^2$ is the coefficient of determination ($R^2 = 1-(SSE/TSS)$); the total sum of squares is denoted as SST; and the residual sum of squares is denoted by SSE. Additionally, $n$ and $p$

are the number of observations or data points and the number of parameters in the equation, respectively. The coefficient of determination is denoted by $R^2$ and is a statistical measure of how well a regression line can approximate real data points.

The importance of testing a regression line in this study to resolve polynomial models renders the determination coefficient a useful statistic. The $R^2$ value is a measure of the relationship between the general deviation around the average trend expressed by the growth curve model, and its value lies between 0 and 1. If the regression line of a model has a coefficient of determination of 1, it is considered a perfect approximation of the real data. Therefore, the predictions of a model are considered satisfactory if the coefficient of determination approaches unity.

The root mean square deviation is a particularly generalized type of standard deviation that may occur among sets of observations and represents the sample standard variation of differences between the values predicted by a model and the observed values (*Wikipedia, 2017*). Individual differences between predicted values and observed values are referred to as residuals. The RMSE was calculated as follows:

$$RMSE = \sqrt{\frac{SSE}{n - p - 1}} \tag{9}$$

where SSE is the residual sum of squares; $n$ is the total number of observations (data points); and $p$ is the number of parameters in the equation. The SSE is the squared sum of residuals, which is the squared sum of the individual differences between values estimated from the model and the observed values. The SSE determines the level of discrepancy between data and a model and is therefore a vital criterion for model selection.

The RMSE value is a major criterion for analysing the predictability and adequacy of the growth curve model of a desired function. Consequently, the growth curve of a function with the smallest error in its RMSE is considered the most adequate model.

The DW is used in regression analyses to detect relationships between values separated from one another by a particular time lag; these relationships are generally referred to as autocorrelations. The DW was employed as discussed above, and the results are discussed below.

The DW lies between the values of 0 and 4; a value close to 2 suggests no autocorrelation; a value trending towards 0 implies a positive autocorrelation; and a value trending towards 4 implies a negative autocorrelation (*Hossein-Zadeh, 2015*). The DW was calculated using the following formula:

$$DW = \frac{\sum_{t}^{n} (e_t - e_{t-1})^2}{\sum_{t=1}^{n} e_t^2} \tag{10}$$

where $e_t$ is the residual at time $t$, and $e_{t-1}$ is the residual at time $t$−1.

Furthermore, AIC was calculated using the following expression (*Hossein-Zadeh, 2015*):

$$AIC = n\ln(SSE/n) + 2p \tag{11}$$

Akaike's information criterion is a good statistic for comparing models or functions of varying complexity due to its ability to modify the SSE for a different range of parametric inputs in a particular model. A smaller AIC value implies greater consistency between the data and the model. The basic use of AIC is in model selection, which has its foundation in information theory and is carried out by implicit comparison of several models. Notably, if all the contrasting models fit the data poorly, AIC selection is affected but not disrupted during usage.

Meanwhile, the BIC (*Lindberg, Schmidt & Walker, 2015*), which is also a model selection criterion, uses the sum of the most likely outcome of data fitting and the selected model but penalizes the logarithm (Log) of the most likely outcome with a term associated with the complexity of the model, as shown in the following equation:

$$BIC = n\ln(SSE/n) + p\ln(n) \tag{12}$$

Similar to AIC values, smaller BIC values from the above expression suggest better consistency between the data and the model.

The model input variables (parameters) were estimated, and statistical analyses were conducted using R software version 2.14.2 (*R Development Core Team, 2010*). The nonlinear and linear models were implemented by employing the nonlinear system (NLS) and linear modelling (LM) procedures. The LM procedure involved the least-squares method, and the NLS procedure involved the Gauss–Newton algorithm.

## Body weight/size ratio

The body weight/size ratio ($K_{wl}$) was calculated as:

$$K_{wl} = W * L^{-1} \tag{13}$$

where $W$ is the body weight (g), and $L$ is the body length (cm).

## Fulton's condition factor

Fulton's condition factor ($K$) was calculated as follows:

$$K = 100\, W * L^{-3} \tag{14}$$

where $K$ is fatness (g/cm$^3$); $W$ is the weight (g); and $L$ is the length (cm). Generally, larger fatness values indicate that the frog is in good shape (*Jin et al., 2015*; *Mozsár et al., 2015*).

The $K_{wl}$ was calculated for each frog to assess its relative condition. The numerical value of the ratio decreases as the condition of the individual improves (*Heikura, 1977*).

# RESULTS

## Growth curves

The body weights and body sizes of the age 1 animals from weeks 1 to 18 are illustrated in Figs. 1A and 2A, which show S-shaped or sigmoidal trends. Additionally, body weights and sizes were assumed to be dependent on age (weeks) according to several growth models for all frogs. The estimated growth curves were typically sigmoidal; in other words, the growth curves were shaped in the form of the letter S (sigmoidal).

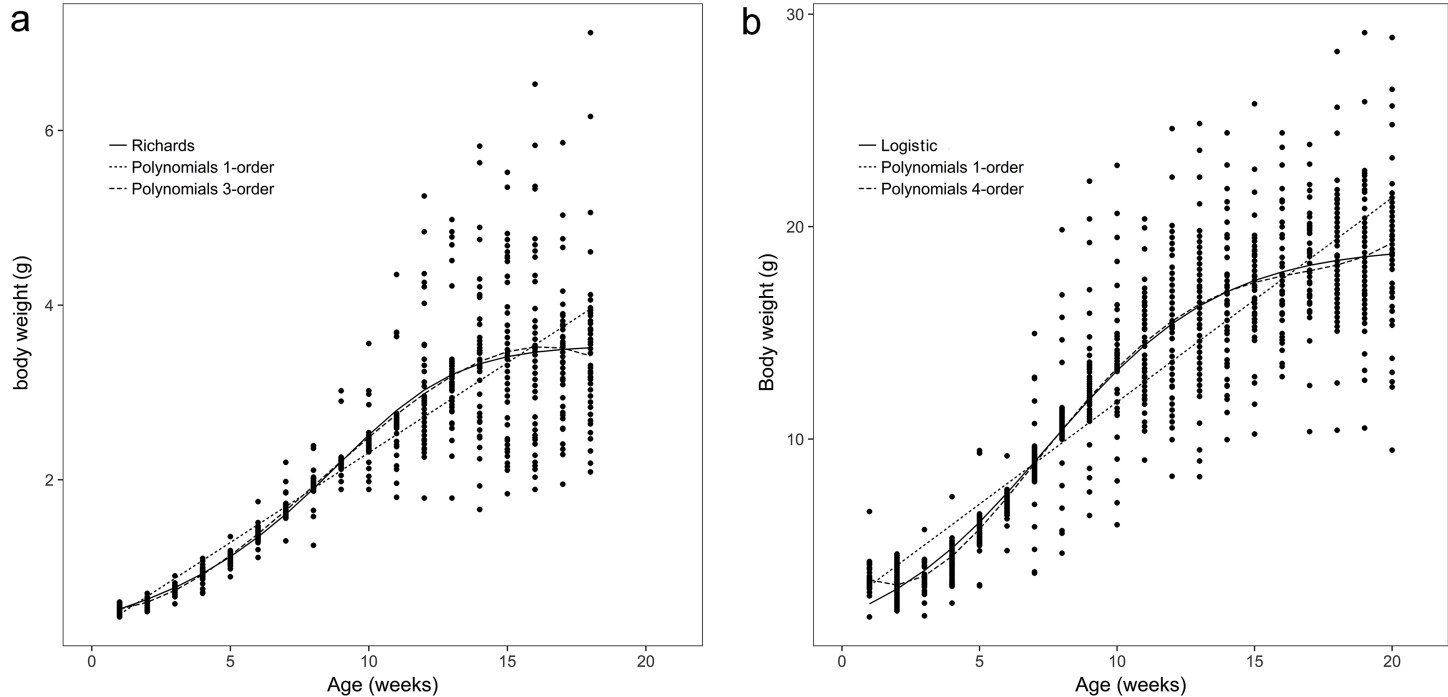

**Figure 1 Observed body weights for brown frogs ages 1 and 2.** (A) The body weight of one-year-old frog within 1–18 weeks. The Polynomial function of third-order was the best in refitting in the equation. Six nonlinear functions was the best fitting in Richards analysis. (B) The body weight of two-year-old frog within 1–20 weeks. The Polynomial function of fourth-order was the best in refitting in the equation. Logistic was the best fitting in six nonlinear functions.

The observed body sizes and shapes of the age 2 frogs from week 1 to week 20 are shown in Figs. 1B and 2B, with an obvious sigmoidal trend in body weights with increasing age. Additionally, the estimated body size (cm) as a function of age (weeks) was examined with several growth models for all frogs, and the original estimated growth curve was sigmoidal.

## Goodness of fit statistics for the growth models

Low AIC, BIC, RMSE, and RMSE values and a high $R^2_{adj}$ value indicate good fit of the model to frog growth, which occurred for the same model.

The results showed that the third- and fourth-order polynomial models provided the most consistent predictions of body weight for age 1 and age 2 brown frogs, respectively, and the Gompertz and third-order polynomial models yielded similarly adequate results for the body size of age 1 brown frogs. Additionally, the Janoschek model generated a similarly adequate result for the body size of two-year-old brown frogs due to decreases in the AIC, BIC, RMSE, and $R^2_{adj}$ values and an increase in the DW value relative to those of the other models (Tables 1 and 2).

Among all the nonlinear functions, the Richards model also provided the most satisfactory results for the body weight of one-year-old frogs. Meanwhile, the most satisfactory model for the body weight of age 2 frogs was the logistic model, and both the Gompertz and the Janoschek models provided the most consistent predictions of

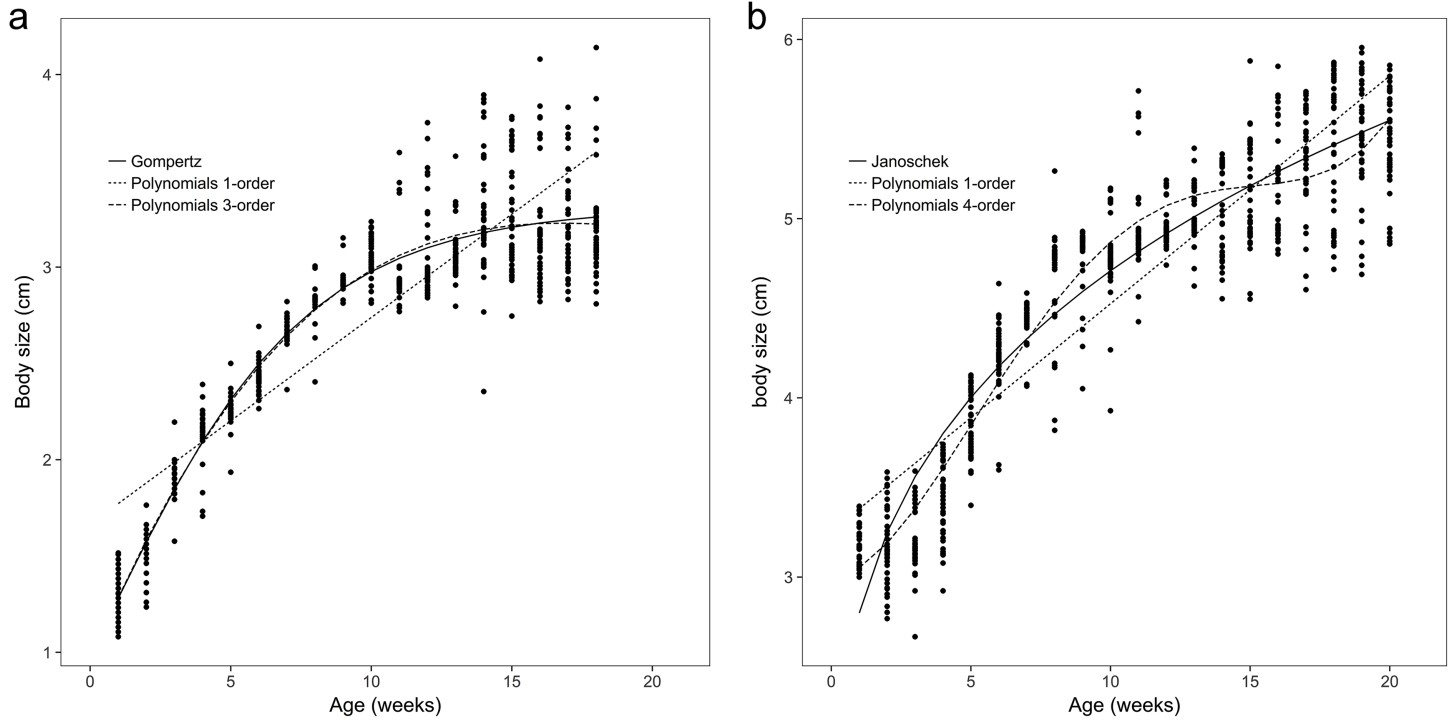

**Figure 2 Observed body sizes for brown frogs ages 1 and 2.** (A) The body length of one-year-old frog within 1–18 weeks. Both the Gompertz and third-order polynomial models yielded similarly adequate results for the body size of age 1 brown frogs. (B) The body length of two-year-old frog within 1–20 weeks. The Janoschek function was the best in refitting in the equation. The Polynomial function of fourth-order was the best in refitting in the Polynomial models.

**Table 1 Comparison of the goodness of fit for different growth curves of body weight in brown frogs.**

| Criterion/model | Age 1 | | | | Age 2 | | | |
|---|---|---|---|---|---|---|---|---|
| | RMSE | DW | AIC | BIC | RMSE | DW | AIC | BIC |
| Logistic | 0.577 | 0.525 | 1,256.063 | 1,274.980 | 2.678 | 0.533 | 3,850.184 | 3,870.923 |
| Gompertz | 0.584 | 0.517 | 1,271.963 | 1,290.280 | 2.705 | 0.526 | 3,866.584 | 3,885.322 |
| Von Bertalanffy | 0.587 | 0.512 | 1,281.243 | 1,299.560 | 2.729 | 0.518 | 3,880.586 | 3,899.324 |
| Brody | 0.598 | 0.497 | 1,307.812 | 1,326.129 | 2.808 | 0.493 | 3,926.194 | 3,944.933 |
| Richards | 0.575 | 0.527 | 1,251.942 | 1,274.838 | 2.679 | 0.533 | 3,851.877 | 3,875.300 |
| Janoschek | 0.599 | 0.482 | 1,310.328 | 1,328.645 | 2.710 | 0.523 | 3,869.361 | 3,888.100 |
| Linear | 0.609 | 0.487 | 1,332.386 | 1,346.124 | 2.955 | 0.449 | 4,007.023 | 4,021.077 |
| Polynomials, second order | 0.596 | 0.499 | 1,302.462 | 1,320.779 | 2.783 | 0.499 | 3,911.700 | 3,930.438 |
| Polynomials, third order | 0.575 | 0.524 | 1,251.922 | 1,274.818 | 2.715 | 0.513 | 3,873.511 | 3,896.934 |
| Polynomials, fourth order | 0.576 | 0.525 | 1,253.712 | 1,281.187 | 2.660 | 0.545 | 3,841.885 | 3,869.992 |

body size for age 1 and 2 frogs, respectively, due to decreases in the AIC, BIC, and RMSE values and increases in the DW and $R^2_{adj}$ values compared to those of the other models (Tables 1 and 2).

The DW of all models was between 0.22 and 0.54. The linear model had the smallest DW values among all considered models of body weight and body size, and the

**Table 2 Comparison of the goodness of fit for different growth curves of body size in brown frogs.**

| Criterion/model | Age 1 | | | | Age 2 | | | |
|---|---|---|---|---|---|---|---|---|
| | RMSE | DW | AIC | BIC | RMSE | DW | AIC | BIC |
| Logistic | 0.192 | 0.508 | −332.067 | −313.750 | 0.269 | 0.382 | 173.674 | 192.412 |
| Gompertz | 0.191 | 0.512 | −333.399 | −315.082 | 0.269 | 0.382 | 173.674 | 192.412 |
| Von Bertalanffy | 0.192 | 0.511 | −330.803 | −312.486 | 0.273 | 0.372 | 196.570 | 215.308 |
| Brody | 0.193 | 0.505 | −319.451 | −301.134 | 0.276 | 0.366 | 212.456 | 231.195 |
| Richards | 0.191 | 0.511 | −332.305 | −309.409 | 0.267 | 0.386 | 161.459 | 184.882 |
| Janoschek | 0.192 | 0.512 | −332.716 | −314.399 | 0.256 | 0.398 | 96.460 | 115.199 |
| Linear | 0.299 | 0.217 | 308.759 | 322.497 | 0.354 | 0.220 | 611.641 | 625.695 |
| Polynomials, second order | 0.194 | 0.489 | −317.717 | −299.400 | 0.274 | 0.367 | 200.948 | 219.687 |
| Polynomials, third order | 0.191 | 0.509 | −333.403 | −310.907 | 0.273 | 0.372 | 197.669 | 221.092 |
| Polynomials, fourth order | 0.191 | 0.509 | −332.389 | −304.913 | 0.261 | 0.408 | 125.982 | 154.089 |

**Table 3 Estimated parameters, $R^2$, and $R^2_{adj}$ for nonlinear growth curve model of body weight in brown frogs.**

| Groups | Criterion/model | Model parameter | | | | $R^2$ | $R^2_{adj}$ |
|---|---|---|---|---|---|---|---|
| | | A | B | C | m | | |
| Age 1 | Logistic | 3.734 | 10.640 | 0.309 | | 0.780 | 0.780 |
| | Gompertz | 4.136 | 1.102 | 0.178 | | 0.775 | 0.775 |
| | Von Bertalanffy | 4.466 | 0.679 | 0.134 | | 0.772 | 0.772 |
| | Brody | 7.058 | 1.007 | 0.043 | | 0.764 | 0.763 |
| | Richards | 3.540 | 322.379 | 0.534 | 2.736 | 0.782 | 0.781 |
| | Janoschek | 3.477 | | 2.656E-3 | 3.542 | 0.763 | 0.762 |
| Age 2 | Logistic | 19.068 | 10.265 | 0.315 | | 0.819 | 0.819 |
| | Gompertz | 20.502 | 1.107 | 0.191 | | 0.815 | 0.815 |
| | Von Bertalanffy | 21.620 | 0.685 | 0.148 | | 0.812 | 0.812 |
| | Brody | 28.703 | 1.015 | 0.060 | | 0.801 | 0.801 |
| | Richards | 18.924 | 15.683 | 0.338 | 1.194 | 0.819 | 0.819 |
| | Janoschek | 18.546 | | 3.572 E-3 | 2.460 | 0.815 | 0.814 |

fourth-order polynomial function presented the largest sets of values with the highest DW values among all considered models of the body weight and body size of age 2 frogs (Tables 1 and 2).

The $R^2$ and $R^2_{adj}$ values of all models showed slight discrepancies between the models of body weight and those of body size. The $R^2_{adj}$ value ranged from 0.762 to 0.822, and the $R^2_{adj}$ values ranged from 0.777 to 0.909. However, the $R^2$ and $R^2_{adj}$ values of body size were higher than those of the models of body weight (Tables 3–6). The Richards model, which is a modification of the sigmoid logistic that allows greater flexibility in S-shaped curves, yielded the greatest $R^2$ and $R^2_{adj}$ values, and the linear model yielded the smallest $R^2$ and $R^2_{adj}$ values for all frogs.

Table 4 Estimated parameters, $R^2$, and $R^2_{adj}$ for the linear and polynomial growth curve models of body weight in brown frogs.

| Criterion/model | Groups | Model parameter | | | | | $R^2$ | $R^2_{adj}$ |
|---|---|---|---|---|---|---|---|---|
| | | $d_0$ | $d_1$ | $d_2$ | $d_3$ | $d_4$ | | |
| Age 1 | Linear | 0.255 | 0.206 | | | | 0.755 | 0.755 |
| | Polynomials, second order | −0.080 | 0.306 | −5.283E-3 | | | 0.766 | 0.765 |
| | Polynomials, third order | 0.504 | −0.019 | 3.640 E-2 | −1462E-3 | | 0.782 | 0.781 |
| | Polynomials, fourth order | 0.555 | −6.304E-2 | 4.611 E-2 | −2.243E-3 | 2.053E-5 | 0.782 | 0.781 |
| Age 2 | Linear | 2.133 | 0.961 | | | | 0.779 | 0.779 |
| | Polynomials, second order | −0.462 | 1.670 | −3.370E-2 | | | 0.805 | 0.804 |
| | Polynomials, third order | 1.736 | 0.548 | 9.660 E-2 | −4.136E-3 | | 0.814 | 0.814 |
| | Polynomials, fourth order | 4.4634 | −1.586 | 0.529 | −3.558 E-2 | 0.748E-3 | 0.822 | 0.821 |

Table 5 Estimated parameters, $R^2$, and $R^2_{adj}$ for the nonlinear growth curve model of body size.

| Groups | Criterion/model | Model parameter | | | | $R^2$ | $R^2_{adj}$ |
|---|---|---|---|---|---|---|---|
| | | $A$ | $B$ | $C$ | $m$ | | |
| Age 1 | Logistic | 3.263 | 2.002 | 0.311 | | 0.909 | 0.908 |
| | Gompertz | 3.310 | 0.191 | 0.244 | | 0.909 | 0.909 |
| | Von Bertalanffy | 3.331 | 0.344 | 0.221 | | 0.908 | 0.908 |
| | Brody | 3.392 | 0.757 | 0.177 | | 0.907 | 0.907 |
| | Richards | 3.290 | 0.532 | 0.268 | 0.367 | 0.909 | 0.909 |
| | Janoschek | 3.302 | | 0.122 | 1.205 | 0.909 | 0.908 |
| Age 2 | Logistic | 5.525 | 1.094 | 0.197 | | 0.891 | 0.891 |
| | Gompertz | 5.628 | −0.253 | 0.156 | | 0.889 | 0.889 |
| | Von Bertalanffy | 5.675 | 0.232 | 0.142 | | 0.888 | 0.888 |
| | Brody | 5.794 | 0.563 | 0.115 | | 0.886 | 0.886 |
| | Richards | 5.390 | 9.331 | 0.302 | 3.436 | 0.893 | 0.893 |
| | Janoschek | 5.315 | | 1.318 E-2 | 2.069 | 0.901 | 0.901 |

## Initial body weights and body sizes

For the linear model, $d_0$ (intercept) is interpreted as an estimate of initial body weights or body sizes, although parameter $d_1$, which is defined as an average body weight or body size, changed over the course of a week. Table 4 shows negative $d_1$ values for the body weight of age 1 frogs, indicating imprecise estimates of this parameter.

## Asymptotic weight and maturation rate

The parameter $A$ is an estimate of asymptotic weight, which can be described as the mature or adult weight. The Brody and Janoschek models yielded the highest and lowest estimates of $A$, respectively, for the body weights of all frogs. The Janoschek growth model is similar to the Richards model in terms of the level of flexibility (Tables 3 and 5).

Another parameter of interest is the speed of growth to the asymptotic weight, which is represented by the parameter $C$ and is called the maturation rate. In this research,

**Table 6 Estimated parameters for the linear and polynomial growth curve models of body size in brown frogs.**

| Criterion/model | Groups | Model parameter | | | | | $R^2$ | $R^2_{adj}$ |
|---|---|---|---|---|---|---|---|---|
| | | $d_0$ | $d_1$ | $d_2$ | $d_3$ | $d_4$ | | |
| Age 1 | Linear | 1.665 | 0.107 | | | | 0.777 | 0.777 |
| | Polynomials, second order | 1.063 | 0.288 | −9.508E-3 | | | 0.907 | 0.907 |
| | Polynomials, third order | 0.951 | 0.351 | −1.754E-2 | 2.819E-3 | | 0.909 | 0.909 |
| | Polynomials, fourth order | 9.220 | 0.375 | −2.293E-2 | 7.148E-4 | −1.139E-5 | 0.909 | 0.909 |
| Age 2 | Linear | 3.254 | 0.127 | | | | 0.812 | 0.811 |
| | Polynomials, second order | 2.671 | 0.286 | −7.571E-3 | | | 0.887 | 0.887 |
| | Polynomials, third order | 2.592 | 0.327 | −1.227E-2 | 1.491E-4 | | 0.888 | 0.888 |
| | Polynomials, fourth order | 2.993 | 1.293E-2 | 5.120E-2 | −4.473E-3 | 1.100E-4 | 0.898 | 0.898 |

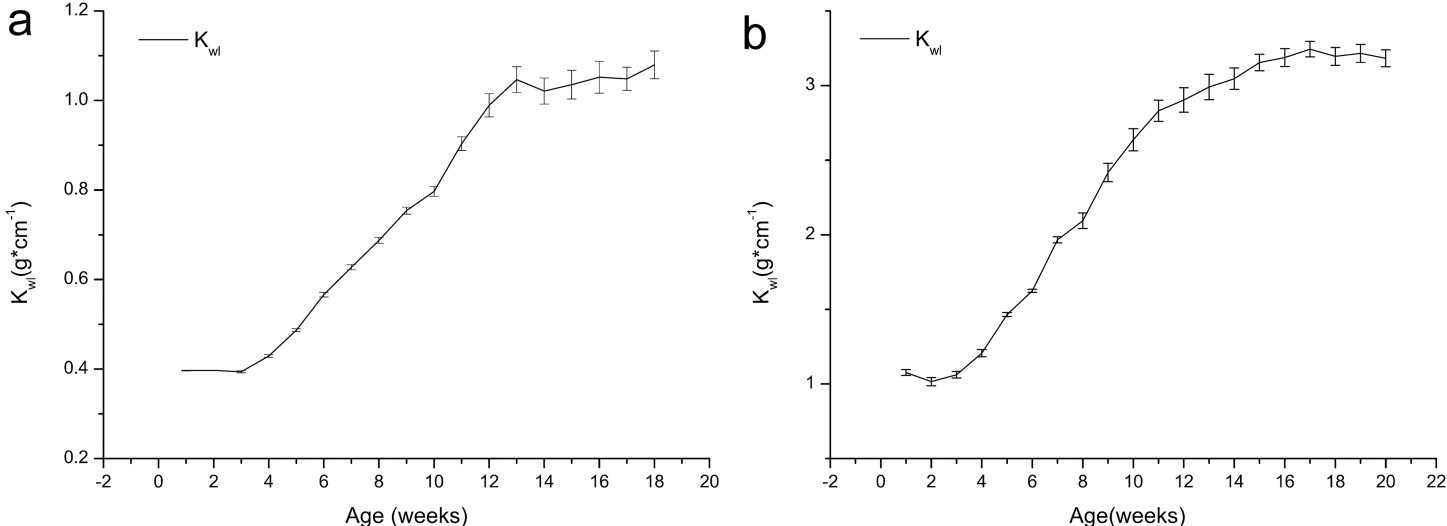

**Figure 3 Variation in the weight/length index ($K_{wl}$) of brown frogs of different ages (mean ± SE).** Body weight/size ratio ($K_{wl}$) was calculated as: $K_{wl} = W^* L^{-1}$, Where $W$ is the weight (g), $L$ is the length (cm). (A) The change of $K_{wl}$ value with the time in one-year-old frog. (B) The change of $K_{wl}$ value with the time in two-year-old frog.

age 2 brown frogs generally showed greater values for this parameter than the age 1 frogs (Tables 3 and 5).

### $K_{wl}$ value and $K$ value

Figure 3 shows that the $K_{wl}$ values of both the age 1 and 2 frogs increased with growth, and no major changes were observed during the first four weeks or from weeks 13 to 18. We can also see that the growth of the frogs was concentrated during weeks 4–14. The $K_{wl}$ value of the one-year-old frogs increased from 0.40 to 1.08, and the $K_{wl}$ value of the age 2 frogs increased from 1.08 to 3.18.

Figure 4 shows that the $K$ value of the one-year-old frogs exhibits uncertain variation, and it decreased from 23.81 to 9.45 in the first four weeks, indicating development of growth characteristics after metamorphosis. The $K$ values increased before declining.

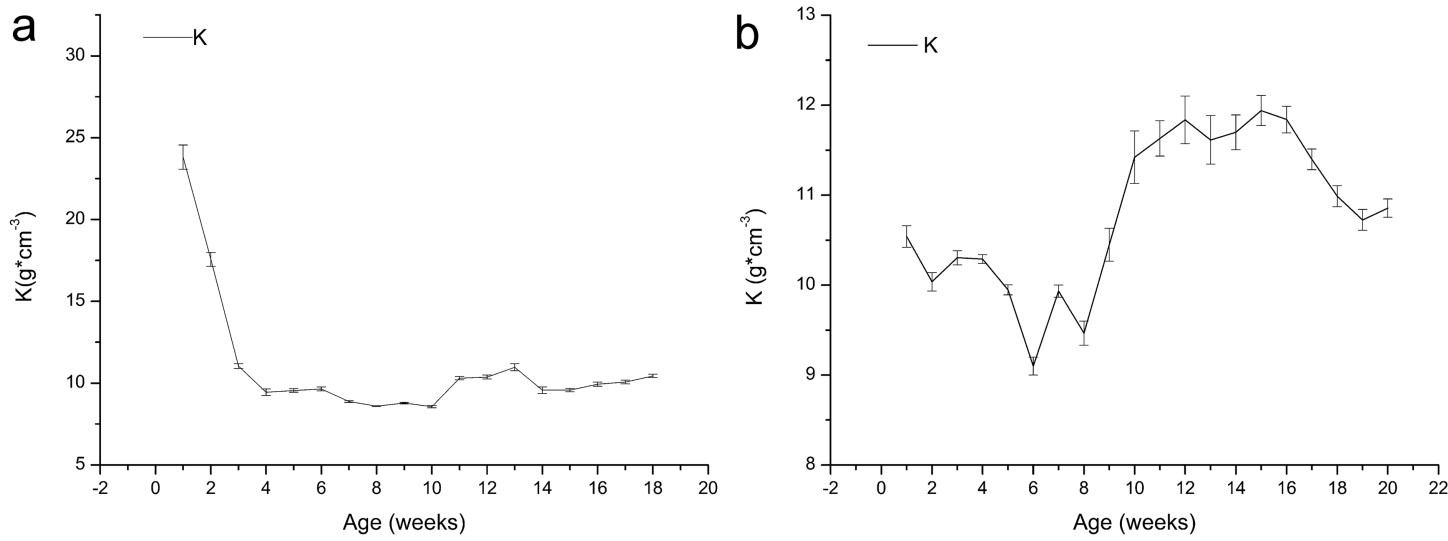

**Figure 4 Variation in the relative fatness of brown frogs of different ages (mean ± SE).** $K = 100 \, W^* L^{-3}$, Where $K$ is fatness (g/cm$^3$), and $W$ is the weight (g), $L$ is the length (cm). Fatness in general larger values, frog is in good shape. (A) The change of $K$ value with the time in one-year-old frog. (B) The change of $K$ value with the time in two-year-old frog.

The $K$ value of the age 2 frogs decreased from 10.54 to 9.10 within six weeks and then increased until week 15 before decreasing again.

## DISCUSSION

### Growth of the brown frogs

The growth of the brown frog, similar to all animals, depends on the genotype-environment interaction (*Mansano et al., 2017a*). Animal growth is associated with genetics, biotype, race, weight, age, and body state. However, culture management and nutrition supply are also the key factors in frog culture (*Mansano et al., 2017b*; *Olvera-Novoa, Ontiveros-Escutia & Flores-Nava, 2007*). Modelling animal growth curves is necessary for optimizing the management and efficiency of animal production (*Köhn, Sharifi & Simianer, 2007*), and growth modelling provides many benefits for frog production. Many similar growth modelling studies of pigs (*Köhn, Sharifi & Simianer, 2007*; *Wellock, Emmans & Kyriazakis, 2004*), sheep (*Hossein-Zadeh, 2015*), poultry (*Rizzi, Contiero & Cassandro, 2013*), fish (*Baer et al., 2011*), crabs (*Durán, Palmer & Pastor, 2013*), and bullfrogs (*Mansano et al., 2012, 2017a*; *Pereira et al., 2014*) are available. *Pereira et al. (2014)* tested Gompertz and logistic models for captive bullfrogs during the fattening phase (the terrestrial phase). In the aquatic phase, different models were applied for evaluation and simulation of bullfrog tadpole growth (*Mansano et al., 2012, 2016*). Monitoring the growth and development of brown frogs is essential to better understand the general relationship between production and management on frog farms (*Pereira et al., 2014*) as well as the relationships between growth and age or growth and feeding over time.

Recent frog studies frequently used the basic unit of year (*López et al., 2017*; *Sarasola-Puente et al., 2011*), and research based on days-old or weeks-old animals are rare

(*Sarasola-Puente et al., 2011*; *Tessa et al., 2017*) even though this scale is very important to studies of froglet growth. However, skeletochronological methods have many deficiencies in terms of studying the growth of amphibians based on days or weeks and many animals must be sacrificed and the work is too difficult and time-intensive.

The artificial breeding technology used for frogs in this study is specific to *R. dybowskii*, which was domesticated in China (*Bing, 2012*). Similar to many fully terrestrial frogs in the temperate zone, *R. dybowskii* lives on land in summer but hibernates in water in the winter. In this study, we built suitable habitats for *R. dybowskii* and investigated its growth in breeding areas. The time points for metamorphosis and out-hibernation and the proximity of the feeding initiation and termination (for hibernation) dates were determined. The ages of the frogs were known from management of the frog farm. The above time points are pertinent to the intrinsic growth rate, which provides a better understanding of growth over weeks or the explicit relationship between growth and feeding.

Studying frog growth and development is essential in the first two years of development (*Iturra-Cid, Ortiz & Ibargüengoytía, 2010*) from metamorphosis to sexual maturity because frogs can reach more than 50% of their maximum body weight during this period (*Yang et al., 2011*). A weight gain of approximately 5–10 times the initial weight was recorded each year. The weights and sizes of the age 1 frogs reflect the growth and development in the current year and affect the next breeding cycle through overwintering survival and slaughter weight for the next year (*Hedeen, 1972*).

The growth curve for frogs is sigmoidal, or S-shaped, from out-hibernation to hibernation, showing a slow rate of increase in the initial stage before accelerating at the metaphase and then slowing again at a later stage (*Hedeen, 1972*; *Martof, 1956*). From the results, we can see that body weight increased throughout the lifetimes of the animals, with only slight decreases in later periods.

The most rapid rate of growth in body weight and size occurs in the first two years (*Iturra-Cid, Ortiz & Ibargüengoytía, 2010*), and the intrinsic growth rates are higher in the first year than those at other times (*Semlitsch, Scott & Pechmann, 1988*). In the first year before hibernation, a weight gain of approximately 7.02-times the initial weight was recorded, and the weight increased during the second year to approximately 5.41-times that of the first year.

As shown in Figs. 1 and 2, growth in terms of body length was faster than that of body weight in the initial stage. In Figs. 3 and 4, the relative fatness index of the brown frogs is low; the growth rates for body weight and body size were unsynchronized, and the growth of body weight was slower than that of body size.

In brown frog breeding, growth requires a reasonable food supply and balanced nutrition, but the frogs often showed dietary deficiencies from the 4th to the 10th week after initiating feeding (*Densmore & Green, 2007*). Therefore, vitamins A and D must be supplied during this period (*Ogilvy, Preziosi & Fidgett, 2012*). Furthermore, growth in terms of body size began prior to the increase in body weight, and growth in terms of body weight started after body size stopped increasing (*Von Bertalanffy, 1957*). The final weight was restricted by body volume and was shown to be self-inhibiting.

Environmental factors, such as temperature (time constraints), obviously restrict growth, as occurs with the gradual decline in temperature before hibernation. The brown frog can experience hypothermia, which directly affects its metabolism (*Ziegler, Arim & Bozinovic, 2016*), followed by a decline in feed intake until feeding and growth stop. Frogs are then faced with six months of hibernation, and they sustain themselves by utilizing energy stored in the liver and fat.

## Making sense of the parameters

Knowledge regarding how parameters, such as the growth rate, differ among all developmental stages in frog breeding is important for formulating culture management strategies. Management must be appropriate for populations at different stages.

In this study, the logistic and Gompertz functions were used because of their ability to simply describe vital factors. Some of these simple descriptions include modelling the growth rate as a basic function of size, continuous growth, asymptotes, parameters with biological points, points of inflection, and sigmoidal trends. According to *Richards (1959)*, the four-parameter flexible inflection points of the Richards function that are suitable for modelling animal growth were developed by modifying the logistic and Gompertz functions (*Richards, 1959*). In contrast, the Janoschek function (*Janoschek, 1957*) can mainly be used to describe the postnatal growth of individuals, but this function is also flexible in terms of its points of inflection, all of which are desirable properties in nonlinear growth models (*Wellock, Emmans & Kyriazakis, 2004*). The fourth-order polynomial model was selected due to its similarity to nonlinear growth models. In this study of brown frogs, the Richards function was found to be most suitable for modelling growth according to the low AIC values, and this conclusion was also determined in a study of the growth of ducks and other poultry (*Knižetova et al., 1991*) after evaluating other studies that used the Richards, logistic, and Gompertz functions to model poultry growth. The $R^2_{adj}$ values for the Richards growth functions were slightly higher than those of the remaining nonlinear models, which all had similar $R^2_{adj}$ values; the Richards growth function also showed the lowest AIC values. The $R^2_{adj}$ value measures linearity and is consequently a more suitable and appropriate factor for linear models.

According to *Tompić et al. (2011)*, the calculated AIC and $R^2_{adj}$ values suggest that the fourth-order polynomial model was more suitable. The estimated $A$ and $C$ parameter values are interpreted as mature body weight and the instantaneous relative growth rate, respectively.

A linear model, unlike nonlinear models, lacks parameters with biological meaning (*Brown, Fitzhugh & Cartwright, 1976*), but nonlinear models with such parameters can be linearized and evaluated by linear regression (*Bates & Hunter, 1985*). The results of the current study suggest that the logistic model is adequate for explaining brown frog growth in terms of both body size and body weight, but further research is required to fully understand the growth of brown frogs in China. Ideally, these studies would involve large frog data sets, each with a similar number of body weight records collected and regulated across intervals that are sufficiently similar to permit sufficient sample sizes.

Obtaining frog descendants with rapid growth and sexual maturity rates (*Dmitriew, 2011*) requires understanding the growth rate, implying that a process exists that alters the curvilinear growth of frogs. There are two strategies for changing growth curves: altering the parameters of a curve while leaving its basic configuration untouched, and changing the basic shape of the curve. The first strategy is the best approach for mature body weight, and the latter strategy is more desirable for the growth rate alone. The second strategy is known to apply to frog culture; the growth in the early stage is exponential, and the frog grows more rapidly if the process is strongly implemented.

## The fitness of the captive environment

*R. dybowskii* is a rare species among breeding frogs. It usually lives on land in a forest environment during its terrestrial stage, which differs from other frogs that exhibit a more typical, amphibious lifecycle.

For *R. dybowskii*, an amphibian that lives in terrestrial habitats, influencing the micro-climate factors of the captive environment was easy, which is unlike cultivation of other frogs or snakes and turtles that have complex lives both in water and on land. The rearing cycle of *R. dybowskii* was up to two- to three-years long, and the frog has the characteristics of 'three adaptions and nine inadaptations'; the three adaptions are gloomy, dampness, and clean, and the nine inadaptations are high light, rain, dry, wind, high temperature, predators, dirty, noise, and off-flavour. These features reflect the high risk of frog culture due to complicated breeding technology and high morality. High-density breeding is the main factor affecting the occurrence of disease and high morality. The present study was conducted at a site within the natural distribution of *R. dybowskii*, and the density of the captive frogs was relatively low, which can largely prevent the occurrence of disease and ensure normal growth of the brown frog.

Many factors affect the growth of frogs, including not only a captive environment but also the quality and quantity of food and health status (*Mansano et al., 2017a*). Culturing frogs in an artificial environment is difficult even if some aspects of the habitat, such as temperature and humidity, satisfy the needs of the animals. For example, the average diurnal temperature recorded inside the facility was $18.73 \pm 3.73$ °C. The temperature of the pens rarely exceeded 35 °C during the trial. If other necessary parameters for optimal culturing are not satisfied, such as density and food composition, growth and health status may change. Typically, animals with free access to food mature faster and grow to larger sizes, and when resource levels decline, animals tend to grow more slowly, reach maturity later, and grow to smaller sizes (*Day & Rowe, 2002*; *Dmitriew, 2011*; *Lind, Persbo & Johansson, 2008*). The breeding system used in this experiment included an appropriate culturing environment, sufficient quality and quantity of food, and proper management, fulfilling the basic requirements for frog growth. The $K$, $K_{wl}$, and fatness of the cultured frogs were greater than those of wild frogs or frogs grown in a much more natural environment (*Wang et al., 1999*), suggesting that frogs can live in a cultured environment and still obtain a not bad nutrition status.

## CONCLUSION

The growth of brown frogs was studied, and a mathematical model was developed using various nonlinear, linear, and polynomial functions. However, third- and fourth-order polynomial models provided the best fit for growth curves of body weight for age 1 and age 2 brown frogs, respectively. Both the Gompertz and third-order polynomial models yielded similarly adequate results for the body size of age 1 brown frogs, while the Janoschek model produced a similarly adequate result for the body size of two-year-old brown frogs due to lower RMSE, AIC, and BIC values and greater $R^2_{adj}$ values than those of the other models. The results will be useful for amphibian research, and the growth curves will inform frog farm management strategies and decision-making regarding the culling of poor producers and the selection of highly productive animals. Furthermore, the relationship between growth characteristics and environmental changes has yet to be revealed.

### Funding

This work was supported by the Heilongjiang Province Natural Science Foundation of China (No. C201046). The funders had no role in study design, data collection and analysis, decision to publish, or preparation of the manuscript.

### Grant Disclosures

The following grant information was disclosed by the authors:
Heilongjiang Province Natural Science Foundation of China: C201046.

### Competing Interests

The authors declare that they have no competing interests.

### Author Contributions

- Qing Tong conceived and designed the experiments, performed the experiments, analysed the data, contributed reagents/materials/analysis tools, prepared figures and/or tables, authored or reviewed drafts of the paper, approved the final draft.
- Xiao-peng Du analysed the data, prepared figures and/or tables, authored or reviewed drafts of the paper, approved the final draft.
- Zong-fu Hu analysed the data, contributed reagents/materials/analysis tools, prepared figures and/or tables, authored or reviewed drafts of the paper, approved the final draft.
- Li-yong Cui performed the experiments, analysed the data, contributed reagents/ materials/analysis tools, authored or reviewed drafts of the paper, approved the final draft.
- Hong-bin Wang conceived and designed the experiments, contributed reagents/ materials/analysis tools, authored or reviewed drafts of the paper, approved the final draft.
## Animal Ethics

The following information was supplied relating to ethical approvals (i.e., approving body and any reference numbers):

All frogs used in this study were handled in strict accordance with Northeast Agricultural University (NEAU) IACUC protocols (IACUC#09-012) and tissues-of-opportunity waivers were approved by NEAU.

## Data Availability

The raw data are provided in a Supplemental File.

## Supplemental Information

Supplemental information for this article can be found online at http://dx.doi.org/10.7717/peerj.4587#supplemental-information.

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
