# Peer review of "Modelling the growth of the brown frog (Rana dybowskii)"

_PeerJ, doi:10.7717/peerj.4587_

## Round 0.1 · original submission · Major Revisions

· Academic Editor

Major Revisions

According to reviewers' comments, a more detailed discussion would be required, besides an improved English language style. You should be aware that some comments are included in the annotated manuscript.

Reviewer 1 ·

Basic reporting

The article is valid and meets the scope of the journal; however before acceptance and publication, it is necessary to improve the manuscript structure. As English is not the mother tongue of the authors, Thus, I suggest improving the language of the article. The literature cited in the article is sufficient to support the hypothesis but I suggest the citation of more studies on growth models in amphibians. The figures should be corrected and if possible reduce the amount of them. The results are interesting; however authors need to improve the discussion.
Suggestions for improving the article are mentioned in the attached file.

Experimental design

The experimental design needs to be improved. In my opinion it is not interesting to produce models for frogs at different ages, since for recommendation of producers and technicians, it will be necessary to apply only one model that corresponds to the whole life of the animal. The application of polynomial and linear models to describe the growth of frogs, I believe is not valid.

Validity of the findings

The findings are valid for publication in the journal, and important for describing the growth of amphibians, since there is very little material for this type of animal production.

Additional comments

The article is valuable and should be published, but I suggest that the authors spend some more time in thoroughly review of the article.
Some basic comments regarding the article structure are mentioned in the attached document.

Annotated reviews are not available for download in order to protect the identity of reviewers who chose to remain anonymous.

Reviewer 2 ·

Basic reporting

I think that authors should read more references of the relationship between body mass/body length and age, especially in Chinese frogs in recent years. For instance, Lu Xin’lab in Wuhan University has published some papers about life-history traits in frogs in recent years. I have added some references related to age-body relationship in the text in attached file.

Experimental design

there are relative small samplings

Validity of the findings

no comment

Additional comments

I have now read and reviewed manuscript entitled “Modelling the growth of the brown frog (Rana dybowskii)”. Authors establish a working model to examine the growth of one-year-old and two-year-old (age 1 and age 2) brown frogs (Rana dybowskii) from metamorphosis or out-hibernation to hibernation under the same environmental conditions. They found that the third and fourth-order polynomial models provided the most consistent predictions of body weight for age 1 and age 2 brown frogs, respectively. Their findings will not only be useful for amphibian research but also for frog farming management strategies and decisions. It is interesting. However, we believe that, if the concerns outlined below can be addressed.
1. I think that authors should read more references of the relationship between body mass/body length and age, especially in Chinese frogs in recent years. For instance, Lu Xin’lab in Wuhan University has published some papers about life-history traits in frogs in recent years. I have added some references related to age-body relationship in the text in attached file.
2. The English expression need improve and some sentences I can not understand means and I can only guess.
3. In the paper, some minor errors in references need corrected and I have marked in text.
Totally, I like this paper and hope that it can be published when my comments have been addressed.
Some questions in attached file nee be also addressed.

Annotated reviews are not available for download in order to protect the identity of reviewers who chose to remain anonymous.

---

## Round 0.2 · Minor Revisions

· Academic Editor

Minor Revisions

Dear authors,

The manuscript comments have been addressed thoroughly. However, the reference section (and references within the text) should be checked as per Reviewer 2. We wait for your reply.

Reviewer 2 ·

Basic reporting

I agree with publishing the paper because it follow my suggestions. There are also some minor questions in references. Please revise carefully them.

Experimental design

no comment

Validity of the findings

no comment

Additional comments

I agree with publishing the paper because it follow my suggestions. There are also some minor questions in references. Please revise carefully them.

Annotated reviews are not available for download in order to protect the identity of reviewers who chose to remain anonymous.

---

## Round 0.3 · accepted · Accept

· Academic Editor

Accept

Dear authors,

Thank you very much for your contribution to PeerJ.

#